# Protocol for Evaluating the Microbial Inactivation of Commercial UV Devices on Plastic Surfaces

**DOI:** 10.3390/mps5040065

**Published:** 2022-07-22

**Authors:** Olivia C. Haley, Yeqi Zhao, Manreet Bhullar

**Affiliations:** 1Department of Horticulture and Natural Resources, Kansas State University, Olathe, KS 66061, USA; ochaley@ksu.edu (O.C.H.); yeqi@ksu.edu (Y.Z.); 2Food Science Institute, Kansas State University, Manhattan, KS 66506, USA

**Keywords:** UVC, *Escherichia coli*, *Staphylococcus aureus*, microbial inactivation

## Abstract

With the plethora of commercially available UV-C devices exhibiting different intensity and lifespans, it is critical to consumer safety that companies verify and clearly communicate the efficacy of their devices as per the intended use. The purpose of this study was to define a low-cost protocol for investigating the antimicrobial efficacy of commercial UV devices for industry use. The tested devices included: a wall-mounted unit (Device A), a troffer unit (Device B), and an induction lamp unit (Device C). The devices were installed within an enclosed tower to prevent the transmission of UV-C radiation outside of the testing area. The procedure details determining the devices′ antimicrobial efficacy using plastic coupons inoculated with *Escherichia coli* or *Staphylococcus aureus.* The protocol includes suggested time–distance treatments according to the potential application of each device type and reports the results as log CFU/mL reduction or percent reduction.

## 1. Introduction

Many human pathogens of high public health interest (i.e., *Influenza*, *E. coli*) are communicable, transmitted from infected to healthy individuals through direct-contact, fomite, airborne, oral, vector, or zoonotic routes. The social and economic impact of the airborne COVID-19 pandemic reignited interest in interventions for air and surface disinfection [1], particularly as the pathogens′ environmental stability confers a high risk of pathogen transmission from aerosolized viral particles and fomites in indoor spaces [2]. Fears and realizations of COVID-19 transmission due to inadequate air recirculation systems and high-touch areas in highly trafficked commercial, industrial, and residential zones led to closures of businesses, schools, and government buildings, resulting in supply chain and lifestyle disruptions [3]. In response to the shortage of conventional antimicrobial interventions (e.g., ethanol, chlorine wipes) in many countries at the beginning of the unprecedented pandemic, the antimicrobial industry shifted its attention towards the research, development, and manufacturing of efficacious, durable, and reusable antimicrobial alternatives.

Ultraviolet (UV) irradiation is one such alternative, having a long-established history of efficacy in the disinfection of non-porous surfaces, air, and water. On the electromagnetic emission spectra, UV light includes wavelengths from 100 to 400 nm, and is further subdivided into: UV-A (315 to 400 nm), UV-B (280 to 315 nm), UV-C (200 to 280 nm), and vacuum UV (100 to 200 nm). UV-C is the most effective range for inactivating microorganisms—hence noted as the ‘germicidal range’ [4,5]—and inactivates microbes by inducing the generation of bipyrimidine photoproducts (i.e., photolesions) [6,7]. For microorganisms unable to compensate for the photodamage through DNA repair mechanisms or upregulation of stress response enzymes, this level of cellular DNA damage results in the cessation of growth and replication, and ultimately cell death [8].

There are a variety of UV-C device geometries for surface and/or air disinfection that have been developed and optimized for industrial or consumer home use. This report examines recently manufactured UV-C devices with four different geometries accommodating a wide of applications: a wall-mounted reactor (Device A), a ceiling-suspended troffer (Device B), and an induction lamp (Device C). The wall-mounted reactor and ceiling-suspended troffer are intended to irradiate an unoccupied area and is increasingly investigated for use in schools [9], medical practices [10,11], and other similar settings. The primary application of the induction lamp is to irradiate the entire cross-section of a heating, ventilation, and air-conditioning (HVAC) duct; ideally, the high UV-C intensity of the induction lamp sterilizes both the duct′s interior surfaces and the air continuously as it circulates through the system [12,13].

The objective of the study was to assess the antimicrobial effectiveness of commercial UV-C systems according to their use and to determine their ability to reduce the risk of communicable disease transmission in their corresponding capacities. *Escherichia coli* (ATCC 25922, Gram-negative) and *Staphylococcus aureus* (ATCC 14458, Gram-positive) were used as test organisms since their UV sensitivity is previously reported [14,15].

From this study, a scientific method protocol was designed to facilitate the further low-cost testing of commercial UV-C devices.

## 2. Experimental Design

This protocol presents the materials and methodology for testing the microbial inactivation efficacy of commercial UV-C devices. The intent of this research group is to provide a low-cost methodology for attaining comparable results across different studies. The general outline of the experiment (shown in Figure 1) includes the following steps: measuring device irradiation, culturing test microorganisms, inactivation procedure, microbial enumeration, and calculations and reporting.

### 2.1. Materials and Reagents

*Escherichia coli* strain (ATCC 25922, provided through the culture collection of the Bhullar Food Safety Lab, Olathe, KS, USA)*Staphylococcus aureus* (ATCC 14458, provided through the culture collection of the Bhullar Food Safety Lab, Olathe, KS, USA)Tryptic Soy Agar (TSA; Thermo Scientific™, Waltham, MA, USA, cat. no. R455004)Tryptic Soy Broth (TSB; Thermo Scientific™, Waltham, MA, USA, cat. no. R455054)Brain–Heart Infusion broth (BHI; Thermo Scientific™, Waltham, MA, USA, cat. no. CM1136B)1X Phosphate Buffer Saline (1X PBS; Thermo Scientific™, Waltham, MA, USA, cat. no. J62036K3)0.1% Buffered Peptone Water (BPW; Thermo Scientific™, Waltham, MA, USA, cat. no. CM0509R)Cell lifter (Corning Incorporated, Corning, NY, USA, cat. no. 3008)Serological Pipette Tip ( ≥30 mL capacity; Thermo Scientific™, Waltham, MA, USA, cat. no. 170357N)Plastic coupons (or petri dish) with a 17.35 cm^2^ surface area (4.7 cm diameter; MilliporeSigma™, Burlington, MA, USA, cat. no. PD2004705)

### 2.2. Equipment

Centrifuge (Beckman-Coulter, Brea, CA, USA; model: Allegra X-30)Optical Light Meter (International Light Technologies, Peabody, MA, USA; model: ILT-2400)254-nm sensor (International Light Technologies, Peabody, MA, USA; model SED005/WBS320/W)Biosafety Cabinet (Labconco Corporation, Kansas City, MO, USA; model: Purifier Logic+ Class II, Type A2)Vortex Mixer (Fisher Scientific, Waltham, MA, USA; model: Fisherbrand™ Analog Vortex Mixer)Serological Pipette (Fisher Scientific, Waltham, MA, USA; model: Fisherbrand™ Electric Pipet Controller)Metal spreader (SP Industries, Inc., Warminster, PA, USA; cat. no. F37736-0009)Turntable (Troemner™, Thorofare, NJ, USA; cat. no. 3040046)

## 3. Procedure

### 3.1. Device Irradiance Measurements

To prepare an enclosed low-cost UV tower with the following dimensions (Figure 1), first, construct the base from wooden beams (or boards). The placement of the horizontal beams should correspond with the experimental distances (0.6 m, 1 m, and 2 m for this protocol).After constructing the base, apply the construction board to the sides using nails or adhesive to cover the testing area. To facilitate the installation of the UV equipment, it may be beneficial to construct a ‘door’ by permanently adhering one side of the construction board to the frame, and using a non-permanent adhesive to affix the other side to the frame.
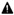
CRITICAL STEP: this step is critical to ensuring human safety in the testing area while the devices are in operation.To install the UV equipment in the tower, remove the construction paper from one side of the tower (or open the ‘door’ if constructed) and fix the equipment onto the horizontal planks at 0.6 m, 1 m, or 2 m using screws. Instead of using a construction board for the top panel, plastic sheeting from a local hardware supplier can be applied to block the transmission of light from the tower.Turn on the UV device and wait for 5 to 10 min before recording the device′s irradiance within the testing area using the ILT-2400 Optical Light Meter equipped with the 254-nm sensor.
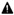
CRITICAL STEP: this warm-up step is critical to ensuring an accurate uniform measurement of UV intensity (μW/cm^2^).OPTIONAL STEP: if interested in reporting the stability of the devices′ UV intensity over time, record the intensity for a 10-min period at each experimental distance.Record and report the UV intensity and dose at each experimental time–distance treatment.

### 3.2. Culturing Test Microorganisms

Propagate Escherichia coli (ATCC 25922) and Staphylococcus aureus (ATCC 14458) by transferring a loopful of frozen culture onto TSA and performing a streak plate.Incubate the inoculated TSA plate at 35 ± 2 °C for 24 ± 2 h; this is the master plate that will be used for the experimental trials.Transfer one colony from the master plate to a culture tube containing 10 mL of TSB or BHI (according to the organism).Incubate the 10 mL of inoculated TSB/BHI at 35 ± 2 °C for 24 ± 2 h.After incubation, vortex the tube and pipette 100 µL of the inoculated medium into a centrifuge tube containing 30 mL of sterile TSB or BHI (according to the organism).Incubate the 30 mL of inoculated TSB/BHI at 35 ± 2 °C for 24 ± 2 h.To prepare the cells for experimentation, centrifuge the tube containing 30 mL of inoculated TSB/BHI at 10,000 rpm and 20 °C for 10 min.Following centrifugation, discard the supernatant and add 30 mL of 1X PBS.Vortex the tube until the pellet is completely dissolved, then centrifuge again at 10,000 rpm and 20 °C for 10 min.Repeat steps 8–9 twice for a total of 3 washes.After the last wash, add 30 mL of 0.1% BPW. This should yield a final concentration of 7–8 log CFU/mL.

### 3.3. Inactivation Experiments

To inoculate a plastic coupon, dispense 100 µL of the washed cell suspension onto the coupon, forming ~50 beads of ~2 µL each in concentric circles (Figure 2).Dry the coupon in a biosafety cabinet for 15 min.
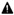
CRITICAL STEP: do not overdry the pellet as this could negatively affect the initial and final microbial population.Activate the UV device for a 10-min warmup period.Place the samples on a plastic tray in a randomized fashion and slide the tray inside the testing area underneath the illuminated UV device (Figure 1). Store untreated control samples outside of the testing area during the experiment.After the treatment time has lapsed, retrieve the plastic tray from the testing area and remove the treated coupons.Immediately place the treated coupons into a Whirlpak bag containing 20mL of 0.1% BPW.
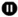
PAUSE STEP: after immersing the treated coupons in 20 mL of 0.1% BPW, the lab recommends processing the samples within one hour to ensure an accurate count.

### 3.4. Microbial Enumeration

Retrieve the Whirlpak bags containing the treated coupons immersed in 20 mL of 0.1% BPW.Using a sterile cell lifter tool, scrape the surface of the plastic coupon for 30 s.
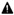
CRITICAL STEP: do not rely solely on hand massaging as the cell lifter tool is required to completely transfer the plaques of bacteria formed during the drying process to the diluent.After scraping the plastic coupon, hand massage the sample bag to homogenize the sample.Perform ten-fold serial dilutions of the sample using 9 mL of 0.1% BPW as the diluent (Steps 5–11).First, prepare a series of sterile culture tubes containing 9 mL of 0.1% BPW. Label the tubes T1, T2, T3, T4, and T5.Add 1 mL of diluent from the homogenized, original sample to T1 and vortex the tube.Add 1 mL of diluent from T1 to T2 and vortex T2.Add 1 mL of diluent from T2 to T3 and vortex T3.Add 1 mL of diluent from T3 to T4 and vortex T4.Add 1 mL of diluent from T4 to T5 and vortex T5.Pipet 100 µL from each tube onto separate TSA plates and spread plate the samples using a flame sterilized spreader. Perform this step in duplicate.Perform steps 4–11 to prepare serial dilutions of the the inoculum to confirm the inoculum′s initial concentration. There should be a minimum of 6 to 7 log CFU/mL present in the inoculum.Using a flame sterilized spreader, spread plate 100 µL of the samples in duplicate onto TSA.Rest the plates for 5 min, then invert the plates and incubate at 35 ± 2 °C for 24 ± 2 h.After incubation, record the counts for plates yielding 25 to 250 colony forming units (CFU) and calculate the log population according to Equation (1):(1)$$log\left(Average\;Count\times 10\times 10^{x}\times 20\right)\;\;$$
where the average count is the average number of colonies observed on the recorded duplicate plates, 10 accounts for the dilution when plating 0.1 mL, x is the plating factor, and 20 represents the volume of diluent added to the sample bag to dilute the original sample. For this protocol, the plating factor is the number of 10-fold serial dilutions prepared times the sample was diluted. For example, the plating factor is 3 if the enumerated plate was prepared with the solution containing 1/1000th the initial concentration of the original sample.Calculate the log reduction and percent reduction according to Equations (2) and (3), respectively.
(2)$$log_{10}\left(\frac{A}{B}\right)$$
where A is the number of viable cells detectable before treatment and B is the number of viable cells detectable after treatment with the UV device.
(3)$$\left(\frac{A-B}{A}\right)\times 100$$
where A is the number of viable cells detectable before treatment and B is the number of viable cells detectable after treatment with the UV device.Of note, either metric may be used to represent the microbial inactivation acheieved with a UV device. However, percent reduction is more common in industry reporting, whereas log reduction is primarily used in the academic (research) setting.
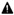
CRITICAL STEP: Repeat the experiment at least twice to estimate the variability of the results and increase the accuracy of your results.

## 4. Expected Results

This protocol should yield inoculated plastic coupons colonized by at least six log CFU of *E. coli* or *S. aureus* with minimal differences between the populations of the control coupons. If there are large differences between the control populations, or a lower concentration of microorganisms observed, investigate the dry time. It is likely that the incubated coupons are being over-dried, which negatively impacts microbial survival. Secondly, investigate the scraping methodology as this step is crucial in establishing a high recovery.

The results of the protocol will vary based on the time–distance combinations used in the trials. For the time–distance combinations recommended for products tested in this protocol, the surviving microbial population should be inversely affected by time and directly proportional to distance. Otherwise, the dry time and scraping methodology should be investigated, as well as potential sources of contamination during the trials.

The data from these experiments can be analyzed in Microsoft Excel (or a similar program) from which the standard deviation can also be computed as a measurement of variability in the data. For reporting purposes, the log reduction, standard deviation, and UV intensity should be reported for each time–distance interval. The reader should also have an indication as to the sample size.

## 5. Reagents Setup

0.1% PBS (1% PBS, distilled water; note that 1X PBS can be stored for several months at ambient room temperature).

## Figures and Tables

**Figure 1 mps-05-00065-f001:**
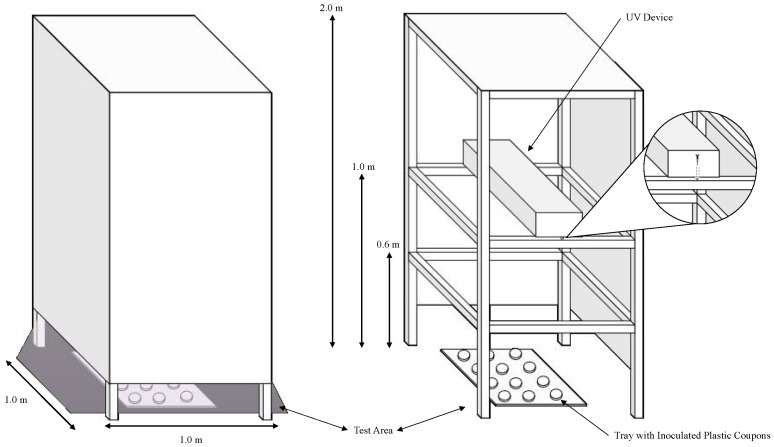
The enclosed UV tower constructed by the lab is intended to block the transmission of UV light outside of the testing area while a test device is in operation. This figure also depicts the completed construction and its intended operation (**left**), the testing area containing the tray with plastic coupons, and how to install the UV device therein (**right**).

**Figure 2 mps-05-00065-f002:**
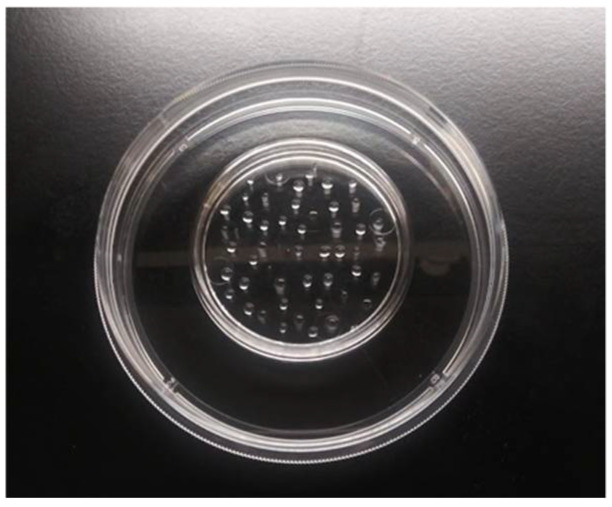
A plastic coupon inoculated with 100 µL of inoculum dispensed into concentric circles of approximately 50 beads containing 2 µL each. Note this photo was taken before the inoculum was dried.

## Data Availability

Not applicable.

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
