# Peer review of "Protocol for Evaluating the Microbial Inactivation of Commercial UV Devices on Plastic Surfaces"

_mps, 2022, doi:10.3390/mps5040065_

Round 1
Reviewer 1 Report
Dear Editor and Authors,
The manuscript ‘Protocol for Evaluating the Microbial Inactivation of Commercial UV Devices on Plastic Surfaces’ by Olivia C. Haley, Yeqi Zhao and Manreet Bhullar describes in detail a protocol for measuring UV-C devices antimicrobial efficacy.
The procedure use plastic coupons inoculated with Escherichia coli or Staphylococcus aureus.
The protocol is described in a detail, literature is collected well, and the manuscript is well written.
I recommend acceptance of the manuscript,
Yours sincerely,
Author Response
Dear Reviewer,
Thank you for reviewing our manuscript “Protocol for Evaluating the Microbial Inactivation of Commercial UV Devices on Plastic Surfaces” for submission to MDPI: Methods and Protocols. We appreciate the time and effort that you have dedicated to providing valuable feedback on our manuscript.
We noted that there were no recommendations for edits to the manuscript, but should you have any follow-up questions please do not hesitate to reach out to us at any time at [email protected].
Sincerely,
Olivia C. Haley
Graduate Research Assistant
Department of Horticulture and Natural Resources
Kansas State University – Olathe
22201 W. Innovation Dr.
Olathe, KS 66061
Tel: +1 913 603 5737
Email: [email protected]

Reviewer 2 Report
The manuscript “Protocol for Evaluating the Microbial Inactivation of Commercial UV Devices on Plastic Surfaces” by Haley et al. reports a method to evaluate the microbial inactivation obtained by exposure to UV lamps on plastic surfaces. The paper is well written but should explain more in depth the procedure.
· It is not clear where to place the UV device and the sample, maybe there should be a scheme to make it clearer, also to understand the distance between them
· The sentence “Prepare serial dilutions using 9mL of 0.1% BPW” could be broken down in order to clarify the dilution process
· The sentence “d represents the amount of diluent added to the sample bag to dilute the original sample” generates some doubts since up to this point all volumes have been clearly determined. If it corresponds to a number, it should be specified, even between brackets.
Author Response
Dear Reviewer,
Thank you for reviewing our manuscript “Protocol for Evaluating the Microbial Inactivation of Commercial UV Devices on Plastic Surfaces” for submission to MDPI: Methods and Protocols.
We appreciate the time and effort that you have dedicated to providing valuable feedback on our manuscript and have been able to incorporate changes to reflect your suggestions. The changes have been highlighted within the manuscript.
Please find below an itemized, numbered response to your comments and concerns:
- Comment: It is not clear where to place the UV device and the sample, maybe there should be a scheme to make it clearer, also to understand the distance between them.
Response: Agreed. The figure was amended to demonstrate how a device could be mounted on the UV tower and how the coupons were placed in the sample treatment area. (Figure 1). More information was also provided in the text regarding the placement of the coupons in the sample area (Lines 210 - 212).
- Comment: The sentence “Prepare serial dilutions using 9mL of 0.1% BPW” could be broken down in order to clarify the dilution process
Response: Agreed. Further clarification was provided (Lines 235 – 245) outlining the dilution process.
- Comment: The sentence “d represents the amount of diluent added to the sample bag to dilute the original sample” generates some doubts since up to this point all volumes have been clearly determined. If it corresponds to a number, it should be specified, even between brackets.
Response: Agreed. Further clarification was provided (Line 257) to specify the value of d specific to this protocol (20).
We look forward to hearing from you in due time regarding our corrections and to responding to any further questions and comments you may have.
Should you have any follow-up questions please do not hesitate to reach out to us at any time at [email protected].
Sincerely,
Olivia C. Haley
Graduate Research Assistant
Department of Horticulture and Natural Resources
Kansas State University – Olathe
22201 W. Innovation Dr.
Olathe, KS 66061
Tel: +1 913 603 5737
Email: [email protected]

Reviewer 3 Report
Development of the method evaluating the microbial inactivation of commercial UV devices on plastic surfaces is needed and the manuscript meets this condition.
In the section Introduction: the authors should explain why they recommend E. coli ATCC 25922 and S. aureus ATCC 14458 strains.
Line 104: Is the shelf at a constant height in the UV tower or can it be repositioned in the cabinet? Where will the UV lamp be installed?
Line 138: Should be step 12: check the number of cells in the inoculum. For example, from a test tube (step 11), transfer of 50 μl of inoculum to 10 ml of 0.1% BPW. Then vortex the tube, and transfer of 1 ml of suspension to a TSA plate. Incubate the TSA plate at 35 ± 2 ºC for 24 ± 2 hours. After incubation, count colonies. There should be over 50 colonies.
Line 183: please correct and add the formula number: Equation 3 respectively.
Line 186: Formula 3 is not clear, for example: (107-103) / 103 x 100 = 101 x 100 = 1000. What does it mean?
Author Response
Dear Reviewer,
Thank you for reviewing our manuscript “Protocol for Evaluating the Microbial Inactivation of Commercial UV Devices on Plastic Surfaces” for submission to MDPI: Methods and Protocols.
We appreciate the time and effort that you have dedicated to providing valuable feedback on our manuscript, and have been able to incorporate changes to reflect your suggestions. The changes have been highlighted within the manuscript.
Please find below an itemized, numbered response to your comments and concerns:
- Comment: In the section Introduction: the authors should explain why they recommend E. coli ATCC 25922 and S. aureus ATCC 14458 strains.
Response: Agreed. Further information was provided regarding the strains’ suitability for antimicrobial studies (Lines 65-67).
- Comment: Line 104: Is the shelf at a constant height in the UV tower or can it be repositioned in the cabinet? Where will the UV lamp be installed?
Response: Thank you for these questions. Further clarification on the UV tower was provided in the figure and text. The figure was amended to demonstrate how a device could be mounted on the UV tower and how the coupons were placed in the sample treatment area. (Figure 1). More information was also provided in the text regarding how to construct the UV tower (Lines 128 – 135), install the devices (Lines 143 - 147) and the placement of the coupons in the testing area (Lines 210 - 211).
- Comment: Line 138: Should be step 12: check the number of cells in the inoculum. For example, from a test tube (step 11), transfer of 50 μl of inoculum to 10 ml of 0.1% BPW. Then vortex the tube, and transfer of 1 ml of suspension to a TSA plate. Incubate the TSA plate at 35 ± 2 ºC for 24 ± 2 hours. After incubation, count colonies. There should be over 50 colonies.
Response: This is a great point. We have added further clarification to this point in the text (Lines 235 – 248) to detailed how to verify the number of viable cells in the culture.
- Comment: Line 183: please correct and add the formula number: Equation 3 respectively.
Response: Agreed. This typo was corrected in the text (Line 269)
- Comment: Line 138: Line 186: Formula 3 is not clear, for example: (107-103) / 103 x 100 = 101 x 100 = 1000.What does it mean?
Response: Agreed. This was a typo; the equation was corrected to ((A – B)/A) x 100) which yields the percent reduction (i.e., 99.9% reduction, 99.999% reduction). Further information was also provided for the context of this measurement (Lines 276 – 279); percent reduction is commonly used in industry reporting whereas log reduction is primarily used in academia.
We look forward to hearing from you in due time regarding our corrections and to responding to any further questions and comments you may have.
Should you have any follow-up questions please do not hesitate to reach out to us at any time at [email protected].
Sincerely,
Olivia C. Haley
Graduate Research Assistant
Department of Horticulture and Natural Resources
Kansas State University – Olathe
22201 W. Innovation Dr.
Olathe, KS 66061
Tel: +1 913 603 5737
Email: [email protected]

Reviewer 4 Report
Protocol for Evaluating the Microbial Inactivation of Commercial UV Devices on Plastic Surfaces is document that meets the requirements for structure and form. It is carefully written and understandable for its content. Numbers of 16 References are used on 7 pages of text, all of them are used purposefully.
For this reason, I can state that I have no comments and express my acceptance of it.
Author Response

(The authors gave the same response as above.)
